# Dietary Patterns and Breast Cancer Risk in Black Urban South African Women: The SABC Study

**DOI:** 10.3390/nu13114106

**Published:** 2021-11-16

**Authors:** Inarie Jacobs, Christine Taljaard-Krugell, Mariaan Wicks, Herbert Cubasch, Maureen Joffe, Ria Laubscher, Isabelle Romieu, Carine Biessy, Sabina Rinaldi, Inge Huybrechts

**Affiliations:** 1Centre of Excellence for Nutrition, North-West University, Potchefstroom 2531, South Africa; christine.taljaard@nwu.ac.za (C.T.-K.); 13009494@nwu.ac.za (M.W.); 2Department of Surgery, Faculty of Health Sciences, University of Witwatersrand, Houghton, Johannesburg 2041, South Africa; herbert.cubasch@wits.ac.za; 3Non-Communicable Diseases Research Division, Wits Health Consortium (PTY) Ltd., Parktown, Johannesburg 2193, South Africa; mjoffe@witshealth.co.za; 4MRC Developmental Pathways to Health Research Unit, Department of Paediatrics, Faculty of Health Sciences, University of Witwatersrand, Johannesburg 2050, South Africa; 5South African Medical Research Council Tygerberg, Cape Town 7505, South Africa; Ria.Laubscher@mrc.ac.za; 6Centro de Investigación en Salud Poblacional, Instituto Nacional de Salud Pública, Cuernavaca 62100, Mexico; iromieu@gmail.com; 7Hubert Department of Global Health, Emory University, Atlanta, GA 30329, USA; 8International Agency for Research on Cancer, Nutrition and Metabolism Branch, IARC-WHO, 150 Cours Albert Thomas, 69372 Lyon, France; biessyc@iarc.fr (C.B.); rinaldis@iarc.fr (S.R.); Huybrechtsi@iarc.fr (I.H.)

**Keywords:** dietary patterns, breast cancer risk, South Africa, black urban women

## Abstract

A total of 396 breast cancer cases and 396 population-based controls from the South African Breast Cancer study (SABC) matched on age and demographic settings was included. Validated questionnaires were used to collect dietary and epidemiological data. Dietary patterns were derived using principal component analysis with a covariance matrix from 33 food groups. Odds ratios and 95% confidence intervals were estimated using conditional logistic regression. A traditional, a cereal-dairy breakfast and a processed food dietary pattern were identified, which together explained 40.3% of the total variance in the diet. After adjusting for potential confounders, the traditional dietary pattern and cereal-dairy breakfast dietary pattern were inversely associated with breast cancer risk (highest tertile versus lowest tertile) (OR = 0.72, 95%CI: 0.57–0.89, *p*-trend = 0.004 and OR = 0.73, 95%CI: 0.59–0.90, *p*-trend = 0.004, respectively). The processed food dietary pattern was not significantly associated with breast cancer risk. The results of this study show that a traditional dietary pattern and a cereal-dairy breakfast dietary pattern may reduce the risk of developing breast cancer in this population.

## 1. Introduction

Diets in South Africa have shifted from nutritious traditional meals towards diets characterized by higher consumption of nutrient-poor, energy-dense foods [1,2,3,4]. Diets comprising more nutrient poor and energy-dense foods have been associated with an increased risk of obesity and other noncommunicable diseases, such as breast cancer [5,6]. Breast cancer is the most frequently diagnosed cancer in South African women, and mortality rates are rapidly increasing [7]. A lack of early cancer screening and costly cancer treatment contribute to high mortality rates in South Africa [8]. Preventing breast cancer is therefore a priority to reduce high incidence rates and the burden on the public healthcare system in South Africa [8].

It is already established that modifiable lifestyle factors such as diet, body weight and physical activity play a crucial role in cancer prevention [9]. However, research on the association between diet and breast cancer risk in South Africa is limited. A previous study conducted in black women from Soweto, South Africa, showed that higher adherence to an adapted version of the 2018 World Cancer Research Fund/American Institute for Cancer Research’s Cancer Prevention Recommendations was inversely associated with breast cancer risk [10]. Another study conducted on black women from Soweto, investigated the association between the degree of food processing and breast cancer risk. In this study, higher intakes of minimally or unprocessed food were inversely associated with breast cancer risk while processed and ultra-processed foods did not show any significant association with breast cancer risk [11]. While these studies mentioned above contribute to valuable insights into the diets of black women form Soweto, more research is required to understand the association between diet and breast cancer risk in South Africa.

There are various different methods to investigate the association between dietary intake and cancer risk, and dietary pattern analysis has emerged as a complementary method over investigating individual nutrients or foods [12], as it allows for an investigation of the effects of overall diets [13]. Dietary patterns can be derived either a priori or a posteriori [14]. The a priori method refers to the use of a scoring system (healthy eating/diet quality index) to calculate adherence to a predefined dietary pattern whereas the a posteriori method (a data-driven approach) refers to the use of statistical modelling techniques such as principle component analysis or exploratory factor analysis to derive dietary patterns empirically [12,13].

Studies investigating a posteriori dietary patterns in association with noncommunicable diseases in the adult South African population are limited. One of these studies showed that dietary patterns comprising predominantly processed foods were positively associated with the risk of being overweight or obese [15]. Obesity is particularly of concern since it is associated with the risk of several noncommunicable diseases, including postmenopausal breast cancer [9]. The association between a posteriori dietary patterns and breast cancer risk in black South African women has not yet been investigated. The aim of this study is, therefore, to determine the association between data-driven dietary patterns and breast cancer risk in black urban women residing in Soweto, South Africa.

## 2. Materials and Methods

### 2.1. Study Population

The subjects included in this study were part of the South African Breast Cancer (SABC) study [16,17,18]. Breast cancer cases were black women and newly diagnosed incidences prior to any cancer treatment from the Chris Hani Baragwanath Academic Hospital. Cases were recruited as soon as possible after the cancer diagnoses. Controls were healthy (not admitted to hospital) black women and unrelated to the breast cancer cases, with no history of cancer diagnoses and matched only by age (±5 years) and area of residence to the cases. Information describing the inclusion and exclusion criteria of breast cancer cases and controls and recruitment of breast cancer cases was previously described elsewhere [16,17]. A total of 396 cases and 396 controls were included in the current analyses.

### 2.2. Personal Information and Lifestyle Status/History

Trained investigators and fieldworkers conducted face-to-face interviews at the time of recruitment using previously validated questionnaires [19,20]. Information regarding socioeconomic and demographics (income, education and other household amenities) were self-reported. Detailed information was further collected with a questionnaire regarding history of health, ethnicity, reproductive risk factors, breast health, family history of cancer, physical activity and smoking habits. Anthropometric measurements such as height, sitting height, weight and waist circumference) were performed according to a standardized protocol, and Body Mass Index (BMI) was calculated as kg/m^2^.

### 2.3. Habitual Dietary Intake

Participants were asked about their habitual dietary intake over the past month, and dietary intake data were collected as soon as possible after breast cancer diagnoses (at recruitment) before any cancer treatment. A validated and culture-specific quantified food frequency questionnaire (QFFQ) was used, together with food models, food portion pictures and household utensils alongside the South African Food Composition Tables to determine habitual dietary intake [21,22,23,24]. A detailed description of the QFFQ and method used to determine the daily intakes are described elsewhere [10]. The nutrient and energy intakes (EI) were calculated by multiplying the daily intake of each food item by the nutrient and energy content (per 100 g), derived from the South African Food Composition Tables, and then by adding the contribution from all food items together [24].

### 2.4. Categorizing of Food Groups to Determine Dietary Patterns

All individual foods and beverages contained in the QFFQ were categorized into 33 food groups (measured in grams/day) based on similarity of the nutrient content (e.g., protein, saturated fat, unsaturated fat, type of carbohydrate, added sugar, fibre or micro-nutrients). Certain individual foods were classified as individual groups on their own since they were consumed often within the population (bread, maize meal, organ/offal meat and peanuts/peanut butter).

### 2.5. Ethical Approval

The International Agency for Research on Cancer and the University of the Witwatersrand Committee for Research on Human Subjects granted ethical approval for the South African Breast Cancer study (M140980). Permission to conduct research at Chris Hani Baragwanath academic hospital was obtained from the Gauteng Province Medical Advisory Committee. All subjects gave written informed consent prior to participation.

### 2.6. Statistical Analysis

Descriptive analyses were performed, and differences between cases and controls were assessed using paired sample t-test (normal distributed data presented as mean ± standard deviation) and Wilcoxon Signed Rank test (not normal data, presented as median, and 25th and 75th percentiles) for continuous variables and paired Chi-square test for categorical variables (presented as percentages). Specifications of the World Health Organization were used to calculate BMI, using measured height and weight (kg/m^2^).

Principal component analysis with a covariance matrix was used to derive a number of independent linear combinations, based on a set of food groups, to retain habitual dietary patterns. This method reduces foods or food groups based on a linear combination of correlated foods or food groups into a smaller set of principle components (dietary patterns) [25]. Although it is preferred that dietary patterns should be uncorrelated, it might be that an individual’s diet consists out of two different patterns at once [26]. For example, a pattern could be characterized by high loadings of vegetables and fruits, together with a pattern characterized by high loadings of refined grains or highly processed foods. For this reason, principle component analysis was the best fit for our data. Normality of food group variables was tested using P–P plots. When food variables were not normally distributed, log-transformation was performed to achieve normality. The Extraction of principle components was followed by orthogonal (varimax) rotation to enhance the interpretability of the dietary patterns [27]. Three components were retained based on a minimum eigenvalue of 1.0, visual inspection of the scree plot, the percentage variance explained and interpretability of the components. Each component was defined by a subset of at least three food groups with an absolute factor loading equal to or greater than −0.21 or 0.21 [27]. If a food group had a factor loading ≥0.21 in more than one pattern, only the one with the highest factor load was considered in the pattern since individuals tend to follow the pattern with the highest score. To validate the suitability of applying principle component analysis on our study sample, the Kaiser Meyer Olkin (KMO) and Bartlett’s Test of Sphericity values were calculated. The obtained KMO value was 0.953 (values close to 1 are considered a very good inter-correlation), and the Bartlett’s test of Sphericity was significant (*p* < 0.001) and indicated homogeneity of variance of the different foods consumed.

### 2.7. Determining the Association between Dietary Patterns and Breast Cancer Risk

Conditional logistic regression models were used to compute odds ratios, and associated 95% confidence intervals were used to determine the association between breast cancer risk and each dietary pattern. Each identified dietary pattern was divided into tertiles based on the 33rd and 66th percentiles of controls to compare the highest to the lowest tertiles to determine the association with breast cancer risk. One standard deviation increase in each dietary pattern (continuous variable) was also used to determine the association with breast cancer risk. Analysis was stratified by hormonal breast cancer receptor subtypes, menopausal status (pre vs. post) and obesity (BMI < 30 kg/m^2^ vs. BMI ≥ 30 kg/m^2^). For the latter two variables, unconditional logistic regression was used.

A three-stage sequential model was used to obtain odds ratios and the associated 95% confidence intervals. Confounding factors were considered factors influencing the crude odds ratios output by more than 10%. The following confounders were examined in the analysis: age (continuous) ethnicity (Zulu/Pedi/Swazi, Xhosa, Sotho, Tshwane, Venda, Tsonga and Ndebele), individual income (R1-R3000, R3001-R6000 and R6001+), level of education (none/primary school, high school and college/postgraduate/diploma), smoking (smokers and non-smokers), height (continuous), waist circumference (continuous), habitual physical activity/d (active and less active), age at menarche (continuous), full-term pregnancy (yes/no), age at first pregnancy (<24 vs. >24 years of age), age at menopause (<48 vs. >48 years of age), time since menopause, parity (≤3 children vs. >3 children), ever breast-feeding (yes/no), duration of exclusive breast-feeding (months), use of exogenous hormones including hormonal birth control to avoid pregnancy (oral contraceptives and injections) and hormone replacement therapy/combined hormone replacement therapy after menopause, family history of breast cancer (yes/no) alcohol consumption, HIV positivity (yes/no), miss-reporting of energy (under reporting vs. over reporting) and total energy intake in kJ (continuous). Only ethnicity, individual income per month, waist circumference, physical activity and menopausal status influenced the crude output by more than 10% and were therefore included in model 2.

Model 3 included all adjustments made in model 2 and additional dietary factors (total energy intake per day, ever alcohol consumption and mutually adjusting for all dietary patterns) to evaluate the additional impact of dietary factors on the association with breast cancer risk in the respective food groups. Sensitivity analysis was conducted by excluding HIV positive breast cancer cases and controls but did not alter the results (results not shown).

## 3. Results

### 3.1. Distribution of Selected Characteristics between Breast Cancer Cases and Controls

Table 1 presents the distribution of selected characteristics between breast cancer cases and control participants. Ethnicity differed significantly among case and control participants with cases having more Ndebele-speaking people and with controls having more Sotho-speaking people. Breast cancer cases had a significant lower waist circumference (93.3 cm ± 13.8 cm) compared with controls (95.8 cm ± 13.7 cm) and had a lower percentage of HIV-positive women (16.5% vs. 22.6%). Considering dietary factors, the percentage of non-alcohol consumers was higher in cases (80.8%) than in controls (69.4%). Additionally, in breast cancer cases, oestrogen positivity (ER+) (75.3%) and progesterone positivity (PR+) (66.4%) were the dominant hormonal breast cancer tumour receptors while triple-negative breast cancer accounted for 16.2% of all tumour types.

### 3.2. Dietary Patterns

Table 2 presents the factor loadings of each retained dietary pattern as well as the percentage variance explained. Three components were retained based on a minimum eigenvalue of 1.0, visual inspection of the scree plot, the percentage variance explained and interpretability of the components. Each component was defined by an absolute factor loading equal or greater than − 0.21 or 0.21. The three components explained 40.3% of the total variance in consumption. Component one, explaining 23.7% of the total variance, predominantly comprised poultry, organ-and-offal meat, mono- and polyunsaturated fats (vegetable oils and margarine), soup powders and vegetables (non-starchy and starchy vegetables) and was named the traditional pattern. Component two explained 9.2% of the total variance and comprised milk, plain yoghurt, unsweetened breakfast cereals, sorghum porridge (oats and maltabella) and fruit juice, while being negatively correlated with maize meal porridge and saturated fats. Component two was named the cereal-dairy breakfast pattern. Component three explained 7.4% of the total variance and comprised cheese, sweetened dairy products, candy/sugar, fast foods, alcoholic beverages, sugar sweetened beverages, fruit spreads or preserved fruits (jam and canned fruit in syrup), and crackers/potato crisps and was named the processed food pattern. Table A1 (Appendix A) presents the nutrient profiles of each dietary pattern per day (comparing the highest tertiles).

The traditional dietary pattern had the lowest total energy content (median = 7356 kJ, 6070 kJ–8925 kJ), followed by the cereal-dairy breakfast pattern (median = 8234 kJ, 6544 kJ–10 931 kJ), and the processed food dietary pattern showed the highest total energy content (median = 12 325, 9589 kJ–15 418 kJ). The processed food dietary pattern had the highest content of saturated fat (median = 27.7 g, 20.3 g–37.1 g) and added sugar (median = 72.8 g, 48.3 g–106.4 g) while showing the lowest content of dietary fibre (mean = 21.7 g ± 8.7 g). The protein-to-carbohydrate-to-fat ratio of each dietary pattern is as follows (calculated as percentages, using each macro-nutrient’s energy content (kJ/d), divided by total energy intake from total protein, carbohydrate, and fat): traditional dietary pattern = 1:5.3:2.8, cereal-dairy breakfast pattern = 1:5.1:2.3 and processed food dietary pattern = 1:4.8:2.5. The processed food dietary pattern also showed the lowest micronutrient content compared with the traditional and cereal-dairy breakfast dietary patterns.

### 3.3. The Association between Dietary Patterns and Breast Cancer Risk

The association between the three retained dietary patterns, comparing the highest with the lowest tertiles of the respective dietary patterns, and breast cancer risk is presented in Table 3. A crude analysis (Model 1) for the traditional dietary pattern showed an inverse association with breast cancer risk overall (OR = 0.76, 95%CI: 0.54–0.94, *p*-trend < 0.001), for postmenopausal women (OR = 0.71, 95%CI: 0.55–0.92, *p*-trend = 0.008), women with PR+ breast cancer (OR = 0.54, 95%CI: 0.34–0.85, *p*-trend = 0.008), for women with a BMI <30 kg/m^2^ (OR = 0.59, 95%CI: 0.45–0.77, *p*-trend < 0.001) and for women with a BMI >30 kg/m^2^ (OR = 0.73, 95%CI: 0.55–0.96, *p*-trend = 0.026). Interestingly, in the fully adjusted model, the inverse association observed in women with BMI > 30 kg/m^2^ lost statistical significance. The crude analysis, comparing the highest to the lowest tertiles of the cereal-dairy breakfast pattern showed inverse associations with breast cancer risk overall (OR = 0.74, 95%CI: 0.62–0.91, *p*-trend = 0.004), for postmenopausal women (OR = 0.76, 95%CI: 0.59–0.97, *p*-trend = 0.027) and for women with a BMI <30 kg/m^2^ (OR = 0.56, 95%CI: 0.43–0.74, *p*-trend < 0.001). Similar inverse associations with breast cancer risk were observed in the fully adjusted model. No statistically significant associations with breast cancer risk were observed for the pattern on processed food. Similar results were observed when dietary patterns were assessed per one standard deviation increase in breast cancer risk (Table A2).

## 4. Discussion

In this black urban population of South African women, a traditional, a cereal-dairy breakfast and a processed food dietary pattern were identified, which together explained 40.3% of the total variance in the diet. After adjusting for potential confounders, the traditional dietary pattern (characterized by poultry, organ-and-offal meat, mono- and polyunsaturated fats, soup powders and vegetables) showed inverse associations with breast cancer risk overall, in postmenopausal women, in women with PR+ breast cancer and in women with a BMI < 30 kg/m^2^. The cereal-dairy breakfast pattern (characterized by milk, plain yoghurt, unsweetened breakfast cereals, sorghum porridge and fruit juice, while being negatively correlated with maize meal porridge and saturated fats) also showed inverse associations with breast cancer risk overall, in postmenopausal women and in women with a BMI < 30 kg/m^2^. No significant association was observed between the processed food dietary pattern (characterized by cheese, sweetened dairy products, candy/sugar, fast foods, alcoholic beverages, sugar sweetened beverages, fruit spreads and crackers/potato crisps) and breast cancer risk.

Numerous case–control and cohort studies, mainly conducted in Caucasian women, have investigated the association between a posteriori defined dietary patterns and breast cancer risk. In general, these studies identified the prudent dietary pattern, characterized by foods such as vegetables, fruit, whole grains, fatty fish, poultry and low fat dairy products, as the dietary pattern is inversely associated with breast cancer risk [25,26,28,29,30,31,32,33,34,35,36,37,38,39].

The a posteriori approach in our study did not identify the same prudent dietary pattern that was observed in previous studies, which also used a posteriori approaches [37,38]. This is probably because our population has many constraints hindering their ability to access and afford a prudent dietary pattern. Different dietary patterns across populations and different study populations under investigation (i.e., black women from low and middle incomes compared with populations from Asia, Europe and America) may further contribute to the different prudent dietary patterns observed in our study. However, the patterns identified in our population, which most resembled the prudent patterns (traditional and cereal-dairy breakfast patterns), were also inversely associated with breast cancer risk. A subcategory analysis of both the traditional dietary pattern and the cereal-dairy breakfast dietary pattern showed inverse associations with breast cancer risk overall, in postmenopausal women, for women with PR+ breast cancer tumours and for women with a BMI < 30 kg/m^2^.

While the amount of foods consumed differed, the traditional dietary pattern in our study contained similar food groups (poultry, vegetables and unprocessed grains) to the prudent dietary patterns identified in other studies using a posteriori approaches. However, the traditional dietary pattern in our study did not contain any fruits while also containing food groups that are not usually included in a prudent dietary pattern, such as organ and offal meat, soup powders, and mono- and polyunsaturated fats such as margarine (excluding fatty fish).

Organ and offal meat are more affordable meat options in South Africa and are often chosen over costlier lean meat cuts, especially red meat, in lower-income households [40]. Organ meat such as the liver can be a good source of protein and certain key micro-nutrients such as iron, which was previously associated with a reduced breast cancer risk in this population [18]. However, organ and offal meat have a higher saturated fat content compared with lean meats [24] and may therefore be considered as less healthy meats in the context of noncommunicable disease prevention. In this population, soup powders and margarine are often used in the preparation of homemade dishes such as meat stews and vegetable dishes or as a sauce eaten together with unprocessed grains. However, due to their high sodium content, these foods are generally considered less healthy foods and are both classified as ultra-processed foods, which have previously been linked to an increased risk for noncommunicable diseases such as breast cancer [41,42].

Of the three dietary patterns, the traditional dietary pattern had the lowest total energy, saturated fat and added sugar content while having the highest amounts for dietary fibre, vitamins and minerals. The traditional diets’ lower energy content indicates that organ/offal meats and margarine were consumed in smaller portion sizes and less frequently. This together with the higher amounts of fibre and micronutrients in the traditional dietary pattern may explain why the traditional dietary pattern was inversely associated with breast cancer risk in this study.

In our study, a cereal-dairy breakfast dietary pattern was inversely associated with breast cancer risk. This may be related to the negative saturated fat loading in this dietary pattern together with the relatively high calcium content of the diet, being the highest of all three identified dietary patterns. Although limited evidence suggests a protective association between diets high in calcium and breast cancer risk, evidence is, however, inconclusive and warrants further investigation [9].

Westernized or unhealthy dietary patterns are often characterized by consumption of fast and deep fried foods, processed meats, saturated fats, sugar sweetened beverages, alcoholic beverages and other highly processed foods. In general, findings from previous studies investigating the association between ‘Westernized’ or unhealthy dietary patterns and breast cancer risk have been inconclusive. For example, a systematic review and meta-analysis, conducted in 2010 and including 17 case–control and cohort studies, did not show any significant association between the highest versus lowest categories of Western/unhealthy dietary patterns (OR = 1.09, 95% CI: 0.98–1.22, *p* = 0.12) [37]. However, a more recent systematic review and meta-analysis conducted in 2019 and including 34 case–control and cohort studies showed a 14% increased risk for developing breast cancer when the highest intake category of the Westernized/prudent dietary pattern was compared with the lowest intake category (OR = 1.14, 95%CI: 1.02–1.28, *p* < 0.001) [38].

In contrast, no significant association between the processed food dietary pattern and breast cancer risk was observed in our study. The results of this study are in line with a former study conducted in black women from Soweto, which investigated the association between ultra-processed food consumption (identified using the NOVA food classification system and breast cancer risk) [11,42]. In the latter study, no significant association was observed between higher ultra-processed food consumption and breast cancer risk [11]. Compared with the highest category of the traditional dietary pattern and the cereal-dairy breakfast pattern, the processed food dietary pattern had the highest total energy, total fat, saturated fat and added sugar content and the lowest fibre and micro-nutrient content. Although such a dietary pattern is not directly associated with breast cancer risk in our population, following a processed food dietary pattern may reduce the overall quality of the diet and may increase the risk of being obese, which is a major risk factor for many chronic diseases and should therefore not be encouraged [9].

In addition, the total variance explained by other studies, investigating a posteriori dietary patterns in association with breast cancer risk, ranged from 10% to 75% [25,26,29,30,31,32,33,34,35,43,44,45]. The total variance explained by dietary patterns in our study (40.3%) was similar to studies conducted in Greece (43%), Argentina (40%), Uruguay (37.8%) and Spain (37%) [25,30,33,43].

The strengths of this study include the fact that cases were recruited prior to any breast cancer treatment, that the questionnaires used to obtain data were proven to be validated, and that the data used in the analysis were standardized and administered by trained personnel. The limitations include the relatively limited sample size of this study; the nature of a case–control study design, which is prone to differential biases of cases; and the use of a QFFQ to collect dietary data, which relies on the memory of participants and is therefore more prone to recall bias. Dietary intake and physical activity were measured over the past month when habitual dietary intake/physical activity of case participants could have changed due to illness and may contribute to random misclassification and under estimation of dietary intake. In addition, although dietary intakes were captured throughout the year (in different participants) seasonal variability of foods (not adjusted for) may have influenced usual reporting of dietary intakes. Ideally large-scale longitudinal studies should confirm the results of this case–control studies that was conducted in the absence of any South African cohort study.

## 5. Conclusions

The results of this study show that a traditional dietary pattern and a cereal-dairy breakfast dietary pattern, consisting of a lower total energy, saturated fat and added sugar and higher fibre, calcium and other key micro-nutrient contents, may reduce the risk of developing breast cancer in this population. Food groups associated with these dietary patterns may play key roles in breast cancer prevention interventions. Following a processed food dietary pattern was not associated with breast cancer risk in our study. However, the higher total energy, saturated fat and added sugar content and lower dietary fibre and key micronutrients content of this processed food dietary pattern may increase the risk of being overweight and obese and ultimately breast cancer risk.

## Figures and Tables

**Table 1 nutrients-13-04106-t001:** Selected characteristics of the study participants by case–control status (means ± standard deviations for parametric data, median and 25th; 75th percentiles for nonparametric data and *n* (%) for categorical variables).

Characteristics	Breast Cancer Cases (*n* = 396)	Controls (*n*= 396)	*p*-Value
**Sociodemographic factors**			
* Age, matched (years)	54.7 ± 12.9	54.6 ± 12.9	0.980
Ethnicity			0.041
Zulu/Pedi/Xhosa/Tswana/Swazi (*n*/%)	67 (16.9)	66 (16.6)	
Sotho (*n*/%)	108 (27.3)	144 (36.4)	
Venda/Tsonga (*n*/%)	105 (26.5)	91 (23.0)	
Ndebele (*n*/%)	116 (29.3)	95 (24.0)	
Level of education			0.078
None/primary (*n*/%)	97 (24.5)	71 (17.9)	
High School (*n*/%)	257 (64.9)	279 (70.5)	
College/University/postgraduate (*n*/%)	42 (10.6)	46 (11.6)	
Individual income/month			0.350
R0 (*n*/%)	125 (31.6)	108 (27.3)	
R1-R3000 (*n*/%)	219 (55.3)	227 (57.3)	
>R3001 (*n*/%)	52 (13.1)	61 (15.4)	
**Anthropometry**			
BMI			0.790
Underweight < 18.5 kg/m^2^ (*n*/%)	5 (1.3)	7 (1.8)	
Normal weight ≥ 18.5 and ≤ 24.9 kg/m^2^ (*n*/%)	63 (15.9)	71 (17.9)	
Overweight ≥ 25.0 and ≤ 29.9 kg/m^2^ (*n*/%)	93 (23.5)	87 (21.9)	
Obese ≥ 30.0 kg/m^2^ (*n*/%)	235 (59.3)	231 (58.4)	
* WC (cm)	93.3 ± 13.8	95.8 ± 13.7	0.011
**Lifestyle factors**			
† Total vigorous and moderate PA min/week	39.4 (7.8; 85.8)	32.1 (9.1; 70.8)	0.303
Current smokers (*n*/%)	35 (8.8)	44 (11.1)	0.286
HIV positivity (*n*/%)	65 (16.4)	90 (22.7)	0.025
**Dietary factors**			
† TE (kJ/day)	9146 (6812; 9759)	8990 (7184; 10,284)	0.239
† Protein (g/day)	63.8 (47.4; 82.7)	63.5 (49.2; 93.1)	0.073
% of TE	11.8	12.0	
† Total fat (g/day)	64.8 (42.4; 91.9)	64.4 (47.2; 95.7)	0.125
% of TE	26.9	27.2	
† Saturated fat (g/day)	17.9 (11.5; 26.1)	19.2 (12.7; 27.9)	0.044
% of TE	7.4	8.1	
* CHO (g/day)	330.8 ± 143.5	338.7 ± 147.3	0.445
% of TE	61.4	64.0	
* Dietary Fibre (g/day)	24.9 ± 11.03	25.3 + 11.4	0.616
† Added Sugar (g/day)	65.3 (38.4; 105.5)	67.9 (39.9; 109.7)	0.313
% of TE	12.1	12.0	
Non-alcohol consumers (*n*%)	350 (88.4)	321 (81.1)	0.004
**Breast cancer risk factors**			
Full term pregnancy in parous women (*n*/%)	377 (95.2)	382 (96.5)	0.374
Ever breast fed in parous women (*n*/%)	339 (91.4)	349 (89.9)	0.293
†‡ Duration of breast feeding (months)	35 (20; 62)	41 (24; 62)	0.187
§ Premenopausal (*n*/%)	133 (33.6)	134 (33.8)	0.852
§ Postmenopausal (*n*/%)	248 (65.1)	257 (65.7)	0.852
† Age at menarche	15 (13; 16)	15 (13; 16)	0.537
†║ Age at menopause (years)	47 (42; 50)	48 (44; 50)	0.331
Family history of breast cancer (*n*/%)	25 (6.3)	17 (4.3)	0.205
Use of birth control (contraceptives) (*n*/%)	229 (57.8)	215 (54.3)	0.316
**Breast cancer case characteristics**			
Receptor status			
ER+ (*n*/%)	298 (75.3)	-	
PR+ (*n*/%)	263 (66.4)	-	
HER2 (*n*/%)	114 (28.8)	-	
¶ Breast Cancer case subtype			
HER2 Enriched (*n*/%)	21 (5.3)	-	
Luminal A (*n*/%)	40 (10.1)	-	
Luminal B (*n*/%)	269 (67.9)	-	
TNBC (*n*/%)	64 (16.2)	-	

WC, waist circumference; TE, total energy; MUFA, monounsaturated fatty acids; PUFA, poly-unsaturated fatty acids; CHO, carbohydrates; PA, physical activity; ER+, oestrogen receptor positive; PR+ progesterone receptor positive; HER2, human epidermal growth factor 2; TNBC, triple-negative breast cancer; HRT, hormone replacement therapy. * Data are presented as means and standard deviations (SD) for parametric data. † Data are presented as median (25th percentile and 75th percentile) for nonparametric data. ‡ In breast feeding women only. § 20 Missing values for menopausal status (15 cases and 5 controls) Missing values were excluded from percentage calculations. ║ Among postmenopausal women only. ¶ Defined using Allred scores.

**Table 2 nutrients-13-04106-t002:** Factor loadings for significant food groups (>0.21) derived from principle component analysis with a covariance matrix and orthogonal rotation applied.

Food Group	Traditional Pattern	Cereal-Dairy Breakfast Pattern	Processed Food Pattern	Unexplained
Milk	−0.0396	**0.4521**	−0.0057	0.5644
Plain yoghurt	0.0135	**0.2354**	0.1825	0.6403
Cheese	0.0866	0.1661	**0.2240**	0.4866
Sweetened milk products	0.0270	0.2084	**0.2568**	0.5416
Eggs	0.1546	0.0543	0.0513	0.7223
Legumes	0.2086	0.0234	−0.1298	0.7339
Red meat	0.1501	0.1027	0.0786	0.5441
Poultry	**0.2746**	−0.0281	−0.1508	0.6313
Fish	0.1671	0.0377	0.0622	0.6882
Organ/offal meat	**0.2803**	−0.1416	−0.0458	0.5721
Processed meat	0.1694	0.1278	−0.0168	0.423
Candy and sugar	0.0829	−0.0031	**0.2327**	0.6087
Sugar Sweetened Beverages	−0.1375	−0.0200	**0.5377**	0.4992
Bread	0.2091	−0.0757	−0.0324	0.7691
Alcoholic beverages	−0.0619	−0.1132	**0.2724**	0.7916
Other drinks *	−0.0705	0.0987	−0.0083	0.9549
Maize Meal porridge	−0.0604	**−0.2823**	0.1336	0.7422
Unprocessed grains †	**0.2194**	0.1367	−0.0338	0.4955
Fast foods ‡	0.1124	−0.0981	**0.3409**	0.5053
Raw Fruits	0.2054	0.1026	−0.0701	0.6305
Fruit juice	−0.0474	**0.2832**	0.0826	0.7903
Fruit spreads/preserved	0.1457	−0.0368	**0.2735**	0.4852
Breakfast Cereals (unsweetened)	0.0784	**0.2860**	0.0154	0.6041
Sorghum porridge (oats and maltabella)	−0.0278	**0.4243**	−0.0744	0.6313
Peanuts and peanut butter	0.1627	0.1291	0.0371	0.6192
Rusks/cookies/sweetened breakfast cereals	0.0782	0.1153	0.1971	0.658
Crackers/potato crisps	0.1447	−0.0274	**0.2465**	0.6389
Mono- and polyunsaturated fats (margarine and vegetable oils)	**0.3377**	−0.0269	−0.0314	0.3313
Saturated fats §	0.0886	**−0.2648**	0.1269	0.8448
Soup powders	**0.3056**	−0.1530	−0.0021	0.4613
Salad dressings and sauces	0.1783	−0.0014	0.1749	0.4604
Starchy vegetables	**0.3203**	−0.0133	−0.0640	0.3526
Non-starchy vegetables	**0.3382**	0.0368	−0.1177	0.2712
Percentage proportion	**23.7%**	**9.2%**	**7.4%**	

* Includes tea and coffee. † Includes rice, pasta, samp and plain popcorn. ‡ Includes pizza, hamburgers, deep fried foods, pies and samosas. § Includes butter, cream, ice-cream, beef tallow and non-dairy coffee creamer.

**Table 3 nutrients-13-04106-t003:** The association between the three retained dietary patterns (highest tertile versus lowest tertile) and breast cancer risk.

		Traditional Dietary Pattern			Cereal-Dairy Breakfast Pattern			Processed Food Pattern		
		OR	95%CI	*p*-Trend §	OR	95%CI	*p*-Trend §	OR	95%CI	*p*-Trend §
Overall (cases *n* = 396; controls *n* = 396)	Model 1	0.76	(0.54–0.94)	<0.00	0.74	(0.62–0.91)	0.004	0.93	(0.76–1.14)	0.494
	Model 2	0.71	(0.57–0.90)	0.003	0.75	(0.62–0.92)	0.006	0.97	(0.78–1.20)	0.785
	Model 3	0.72	(0.57–0.89)	0.004	0.73	(0.59–0.90)	0.004	0.99	(0.79–1.25)	0.964
Premenopausal *,† (*n* = 267) (cases *n* = 133; controls *n* = 134)	Model 1	0.78	(0.54–1.14)	0.206	0.78	(0.56–1.09)	0.149	0.85	(0.59–1.21)	0.375
	Model 2	0.79	(0.54–1.16)	0.242	0.75	(0.53–1.06)	0.105	0.84	(0.59–1.21)	0.364
	Model 3	0.82	(055–1.21)	0.318	0.72	(0.51–1.03)	0.072	0.98	(0.67–1.46)	0.947
Postmenopausal *,† (*n* = 505) (cases *n* = 248; controls *n* = 257)	Model 1	0.71	(0.55–0.92)	0.008	0.76	(0.59–0.97)	0.027	1.01	(0.78–1.29)	0.937
	Model 2	0.74	(0.58–0.96)	0.023	0.79	(0.62–0.98)	0.049	1.01	(0.78–1.29)	0.932
	Model 3	0.73	(0.56–0.93)	0.015	0.78	(0.59–0.98)	0.033	0.97	(0.74–1.28)	0.839
ER+ (*n* = 298)	Model 1	0.76	(0.47–1.24)	0.284	0.96	(0.62–1.49)	0.887	0.77	(0.48–1.23)	0.277
	Model 2	0.84	(0.48–1.49)	0.565	0.82	(0.51–1.39)	0.469	0.84	(0.49–1.43)	0.527
	Model 3	0.86	(0.45–1.66)	0.672	0.83	(0.50–1.37)	0.470	0.96	(0.54–1.72)	0.915
PR+ (*n* = 263)	Model 1	0.54	(0.34–0.85)	0.008	0.72	(0.51–1.04)	0.084	0.78	0.54–1.13)	0.192
	Model 2	0.51	(0.29–0.89)	0.018	0.71	(0.47–1.06)	0.098	0.81	(0.53–1.23)	0.340
	Model 3	0.45	(0.24–0.86)	0.016	0.67	(0.43–1.03)	0.069	0.97	(0.55–1.70)	0.922
BMI <30 kg/m^2^ * (*n* = 326) (cases = 165; controls = 161)	Model 1	0.59	(0.45–0.77)	<0.001	0.56	(0.43–0.74)	<0.001	0.72	(0.54–1.11)	0.387
	Model 2	0.56	(0.42–0.75)	<0.001	0.54	(0.41–0.72)	<0.001	0.69	(0.52–1.09)	0.121
	Model 3	0.41	(0.21–0.77)	0.006	0.33	(0.16–0.64)	0.001	0.98	(0.69–1.39)	0.932
BMI ≥30 kg/m^2^ * (*n* = 466) (cases = 231; controls = 235)	Model 1	0.73	(0.55–0.96)	0.026	0.87	(0.67–1.12)	0.286	1.02)	(0.79–1.33)	0.879
	Model 2	0.74	(0.56–0.99)	0.043	0.91	(0.71–1.18)	0.503	1.00	(0.76–1.31)	0.983
	Model 3	0.75	(0.57–1.01)	0.052	0.90	(0.69–1.17)	0.449	1.10	(0.78–1.41)	0.741

ER+, oestrogen receptor positive; PR+, progesterone receptor positive. Model 1: crude output. Model 2: ethnicity, waist circumference (not adjusted for waist circumference when stratified by obesity, physical activity and menopausal status (not adjusted for menopausal status when stratified by menopausal status)), and under- and over-reporting. Model 3: adjusted for all in Model 2 and adjusted for total energy, ever alcohol intake and mutually adjusted for all dietary patterns. * Unconditional logistic regression. † Twenty missing values for menopausal status (fifteen cases and five controls). § *p*-value for trend analysis of highest versus lowest categories in each model.

## Data Availability

The data presented in this study are available from the corresponding author upon request. The data are not publicly available since access to the SABC data is subject to the approval of the SABC Steering Committee.

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
