# Peer review of "Dietary Patterns and Breast Cancer Risk in Black Urban South African Women: The SABC Study"

_nutrients, 2021, doi:10.3390/nu13114106_

Round 1
Reviewer 1 Report
In the present manuscript, the authors conducted an association study of breast cancer and dietary pattern. The authors identified a traditional, a cereal-dairy breakfast and a processed food dietary pattern. After statistically analysis, the authors found a traditional dietary pattern and a cereal-dairy breakfast dietary pattern were inversely associated with breast cancer risk, therefore concluded these two dietary patterns may reduce the risk of developing breast population. This manuscript was well-thought, and it could have a significant impact of the field. There are some minor concerns that should be addressed before accepting for publication.
- What is the rational of using the principal component analysis method? Please include this in Section 2.6.
- Please include “protein:carbohydrate:fat ratio” for each dietary pattern in Section 3 so that it’s clearer to compare for readers.
- Please keep the format consist through the text. For example, line 241 had extra space before “0.71”; the unit of BMI should be “kg/m2” not “kg/m2”; line 245 lost a space before “>30”; line 315 missed a period sign before “However”; etc.
Author Response
2) What is the rational of using the principal component analysis method? Please include this in Section 2.6.
Thank you for your comment. The following rational for using the principle component analyses was inserted in Section 2.6 (lines 131-137, page 3):
“This method reduces foods or food groups based on a linear combination of correlated foods or food groups into a smaller set of principle components (dietary patterns) (25). Although it is preferred that dietary patterns should be uncorrelated, it might be that an individual’s diet consists out of two different patterns at once (26). For example, a pattern characterized by high loadings of vegetables and fruits, together with a pattern characterized by high loadings of refined grains or highly processed foods. For this reason, principle component analysis was the best fit for our data.”
Please see below the rationale for investigating posteriori dietary patterns in our population (not included in the revised paper):
As foods are not eaten alone but in combination, more and more interest is rising to study the association of different dietary patterns, rather than single foods alone, with cancer risk, including breast cancer (Catsburg et al., 2015; Hu et al., 2002). Probably the most important are the influences dietary patterns have on body weight, which is a known risk factors for postmenopausal breast cancer (Wild et al., 2020). Dietary patterns can be derived using either a priori or a posteriori methods. For the a priori method, a scoring system is used to determine the quality of the diet based on adherence to a predefined dietary pattern (Hu et al., 2002). On the other hand, to determine a posteriori dietary patterns, empirical statistical techniques such as principle component or factor analysis are used and is often referred to as a data-driven approach (Hu et al., 2002; Newby et al., 2004). The latter approach has an advantage over a priori approaches as it considers many aspects of the diet rather than focusing on individual or hypothesized foods or food groups (Nettleton et al., 2008).
Several a posteriori dietary patterns have been explored in association with breast cancer and include Westernized dietary patterns (characterized by frequent consumption of fast-and-deep fried foods, processed meats, saturated fats, sugar sweetened beverages and other highly processed foods) and Mediterranean and/or prudent dietary patterns (characterized by frequent consumption of vegetables, fruit, whole grains, fatty fish, olive oil, poultry and low fat dairy products) (Cottet et al., 2009; Tumas et al., 2014; Castello et al., 2014; Catsburg et al., 2015; Link et al., 2013). Generally, following a Westernized dietary pattern is associated with an increased breast cancer risk while following a Mediterranean or prudent dietary pattern is associated with a reduced breast cancer risk (Dandamudi et al., 2018; Xiao et al., 2019). But, evidence remain inconclusive (WCRF/AICRF, 2018b). There is however, strong evidence that following a Westernized dietary pattern probably increases one’s risk of gaining weight/being obese, which in turn, may increases postmenopausal breast cancer risk (WCRF/AICR, 2018a).
2) Please include “protein:carbohydrate:fat ratio” for each dietary pattern in Section 3 so that it’s clearer to compare for readers.
The following was included in section 3 (lines 241-245, page 8)
The protein: carbohydrate: fat ratio of each dietary pattern is as follows (calculated as percentages, using each macro-nutrient’s energy content (kJ/d), divided by total energy intake from total protein, carbohydrate and fat): traditional dietary pattern=11: 58: 31 (1: 5.3: 2.8) cereal-dairy breakfast pattern=12: 61: 27 (1: 5.1: 2.3) and processed food dietary pattern=12: 58: 30 (1: 4.8: 2.5).
The ratios were also added in Table A1.
3) Please keep the format consist through the text. For example, line 241 had extra space before “0.71”; the unit of BMI should be “kg/m2” not “kg/m2”; line 245 lost a space before “>30”; line 315 missed a period sign before “However”; etc.
Thank you for highlighting these text errors. These have been corrected and the text throughout the paper was checked for errors.
Reviewer 2 Report
This study was designed to explore the relation between dietary patterns and breast cancer risk in black urban South African women. It is found that traditional dietary pattern and a cereal-dairy breakfast dietary pattern may reduce the risk of developing breast cancer in this population. This study was well designed and the manuscript was well prepared. It provides useful information. The weakness is the relative small sample size.
Minor issue:
Discussion section is too long, which should be shortened and focused on the main findings of this study.
Author Response
1) Discussion section is too long, which should be shortened and focused on the main findings of this study.
Thank you for your suggestion. The discussion was reduced and focuses on the main findings of the study now.
Please note that we were asked to include information regarding the total variance explained for dietary pattern by other studies and how it compares to our study. However, this paragraph was written to be specific and condensed to keep the discussion shorter.
Reviewer 3 Report
This manuscript investigates a posteriori dietary pattern and breast cancer risk in South Africa.
Overall the study appears well conducted, analyzed and presented.
A few improvements are however needed.
The inclusion/exclusion criteria for cases an controls should be better described. It appears that the controls were admitted to hospital? What were the reasons for admittance ? Were the limits on age, residence etc.?
The identified patterns explained 40% of total variance. This seems relatively low to me, as compared to other studies. Can the authors compare the % of explained variance in their study to that in the other studies ( eg. those from the systematic reviews they cite) and discuss this point ?
What is the association of the three identified dietary pattern with BMI, social class and other main risk factors for breast cancer ?
Author Response
1) The inclusion/exclusion criteria for cases an controls should be better described. It appears that the controls were admitted to hospital? What were the reasons for admittance ? Were the limits on age, residence etc.?
The control participants were not admitted to hospital. The sentence was revised to reduce confusion (please see lines 83-85, page 2). “Controls were healthy (not admitted to hospital) black women and unrelated to the breast cancer cases with no history of cancer diagnoses and matched only by age (± 5 years) and area of residence to the cases."
More detail regarding the recruitment, inclusion and exclusion criteria was previously described in a paper published by Romieu et al., 2021. A sentence stating the above was added in line 85-87 (page 2).
The following was inserted: “Information describing the inclusion and exclusion criteria of breast cancer cases and controls and recruitment of breast cancer cases was previously described elsewhere [16,17].”
2) The identified patterns explained 40% of total variance. This seems relatively low to me, as compared to other studies. Can the authors compare the % of explained variance in their study to that in the other studies ( eg. those from the systematic reviews they cite) and discuss this point ?
Thank you for your comment. We agree that our % variance explained might be low compared to other studies investigating dietary patterns in general. When we compare our % variance to other studies also investigating the association between a posteriori dietary patterns and breast cancer risk, we observe that our % variance is not as low.
For example, the total variance explained by other studies, investigating a posteriori dietary patterns in association with breast cancer risk, range from 10% to 75% (Buck et al., 2011; Cotett et al., 2009; Ronco et al., 2006;), while the majority of studies reported values between 23% and 43% (De Stefani et al., 2008; Bessaoud et al., 2012; Demitriou et al., 2012; Morouti et al., 2015; Shin et al., 2016; Karimi et al., 2013; Castello et al., 2014; Tumas et al., 2014; Link et al., 2013).
The total variance explained by dietary patterns in our study (40.3%) were similar to studies conducted in Greece (43%), Argentina (40%), Uruguay (37.8%) and Spain (37%) (Mourouti et al., 2015; Tumas et al., 2014; De Stefani et al., 2008; Castello et al., 2014). The differences in total variance explained may be due to the different a posteriori approaches used between studies (factor analysis v. principal component analysis), different number of dietary patterns retained and different populations, for which diets may differ, under investigation.
We have added a sentence in the discussion to indicate that our % variance are similar to other studies in the field. Please see the following: (lines 362-366, page 10):
"In addition, the total variance explained by other studies, investigating a posteriori dietary patterns in association with breast cancer risk, ranged from 10% to 75% [25,26,29,30-35,43-45]. The total variance explained by dietary patterns in our study (40.3%) were similar to studies conducted in Greece (43%), Argentina (40%), Uruguay (37.8%) and Spain (37%) [25,30,33,43]."
3) What is the association of the three identified dietary pattern with BMI, social class and other main risk factors for breast cancer ?
Thank you for your comment. Our paper’s main focus is on the association between dietary patterns and breast cancer risk and did therefore not include other possible breast cancer risk factors in this study. We also decided not to investigate other possible risk factors (or any other outcomes, other than breast cancer) in this population to reduce the complexity and length of the paper. In addition, this paper forms part of a PhD thesis, which mainly investigate the association between diet and breast cancer as the only outcome.
We did however, stratify the three identified dietary patterns according to menopausal-, hormone receptor positive- and BMI status. These analyses were done as dietary risk factors may differ according to menopausal-, hormone receptor positive- and BMI status. It is also worth mentioning that we did adjust for potential confounders/social factors such as ethnicity, individual income per month, waist circumference, physical activity and menopausal status in our regression models. Please see section 2.7 for the list of confounders we tested for in our regression models:
Confounding factors were considered as factors influencing the crude odds ratios output by more than 10%. The following confounders were examined in the analysis: age (continuous) ethnicity (Zulu/Pedi/Swazi, Xhosa, Sotho, Tshwane, Venda, Tsonga and Ndebele), individual income (R1-R3000, R3001-R6000 and R6001+), level of education (none/primary school, high school and college/postgraduate/diploma), smoking (smokers and non-smokers), height (continuous), waist circumference (continuous), habitual physical activity/d (active and less active), age at menarche (continuous), full-term pregnancy (yes/no), age at first pregnancy (<24 v. >24 years of age), age at menopause (<48 v. >48 years of age), time since menopause, parity (≤3 children v. >3 children), ever breast-feeding (yes/no), duration of exclusive breast-feeding (months), use of exogenous hormones including hormonal birth control to avoid pregnancy (oral contraceptives and injections) and hormone replacement therapy/combined hormone replacement therapy after menopause, family history of breast cancer (yes/no) alcohol consumption, HIV positivity (yes/no), miss-reporting of energy (under reporting vs over reporting) and total energy intake in kJ (continuous).
Please see section 2.7. (Analysis was stratified by hormonal breast cancer receptor subtypes, menopausal status (pre vs post) and obesity (BMI <30 kg/m2 vs BMI ≥30 kg/m2).
Please also see section 3.3 and Table 3 where the association between the three retained dietary patterns, comparing the highest to the lowest tertile of the respective dietary patterns, and breast cancer risk is presented.
The following is stated in section 3.3:
“Crude analysis (Model 1) for the traditional dietary pattern showed an inverse association with breast cancer risk overall (OR=0.76, 95%CI: 0.54-0.94, p-trend<0.001), for postmenopausal women (OR=0 .71, 95%CI: 0.55-0.92, p-trend=0.008), women with PR+ breast cancer (OR=0.54, 95%CI: 0.34-0.85, p-trend=0.008), for women with a BMI <30 kg/m2 (OR=0.59, 95%CI: 0.45-0.77, p-trend<0.001) and for women with a BMI >30 kg/m2 (OR=0.73, 95%CI: 0.55-0.96, p-trend=0.026). Interestingly, in the fully adjusted model, the inverse association observed in women with BMI >30 kg/m2 lost statistical significance.